# Bioactivity-Guided Screening of Antimicrobial Secondary Metabolites from Antarctic Cultivable Fungus *Acrostalagmus luteoalbus* CH-6 Combined with Molecular Networking

**DOI:** 10.3390/md20050334

**Published:** 2022-05-19

**Authors:** Ting Shi, Xiang-Qian Li, Ze-Min Wang, Li Zheng, Yan-Yan Yu, Jia-Jia Dai, Da-Yong Shi

**Affiliations:** 1College of Chemical and Biological Engineering, Shandong University of Science and Technology, Qingdao 266590, China; shiting_jia@126.com; 2State Key Laboratory of Microbial Technology, Institute of Microbial Technology, Shandong University, Qingdao 266200, China; lixiangqian@sdu.edu.cn (X.-Q.L.); zemin3210@163.com (Z.-M.W.); yuyanyan@sdu.edu.cn (Y.-Y.Y.); daijiajia@sdu.edu.cn (J.-J.D.); 3Laboratory for Marine Drugs and Bioproducts of Qingdao National Laboratory for Marine Science and Technology, Qingdao 266071, China; 4Key Laboratory of Marine Eco-Environmental Science and Technology, First Institute of Oceanography, Ministry of Natural Resources, Qingdao 266061, China; zhengli@fio.org.cn; 5Laboratory for Marine Ecology and Environmental Science, Qingdao Pilot National Laboratory for Marine Science and Technology, Qingdao 266071, China

**Keywords:** Antarctic fungi, bioactivity-guided screening, molecular networking, antimicrobial activities, secondary metabolites, *Acrostalagmus luteoalbus*, indole diketopiperazines, *α*-pyrones

## Abstract

With the increasingly serious antimicrobial resistance, discovering novel antibiotics has grown impendency. The Antarctic abundant microbial resources, especially fungi, can produce unique bioactive compounds for adapting to the hostile environment. In this study, three Antarctic fungi, *Chrysosporium* sp. HSXSD-11-1, *Cladosporium* sp. HSXSD-12 and *Acrostalagmus luteoalbus* CH-6, were found to have the potential to produce antimicrobial compounds. Furthermore, the crude extracts of CH-6 displayed the strongest antimicrobial activities with 72.3–84.8% growth inhibition against *C. albicans* and *Aeromonas salmonicida*. The secondary metabolites of CH-6 were researched by bioactivity tracking combined with molecular networking and led to the isolation of two new *α*-pyrones, acrostalapyrones A (**1**) and B (**2**), along with one known analog (**3**), and three known indole diketopiperazines (**4**–**6**). The absolute configurations of **1** and **2** were identified through modified Mosher’s method. Compounds **4** and **6** showed strong antimicrobial activities. Remarkably, the antibacterial activity of **6** against *A. salmonicida* displayed two times higher than that of the positive drug Ciprofloxacin. This is the first report to discover *α*-pyrones from the genus *Acrostalagmus*, and the significant antimicrobial activities of **4** and **6** against *C. albicans* and *A. salmonicida*. This study further demonstrates the great potential of Antarctic fungi in the development of new compounds and antibiotics.

## 1. Introduction

Antimicrobial drugs, including antibiotics, antivirals, antifungals, and antimalarials, are medicines that are active against various infections [1]. However, antimicrobial resistance (AMR) has become an increasing state when microorganisms develop resistance to antimicrobial drugs, leading to invalid treatment with existing drugs [2]. AMR is a global concern as new resistance mechanisms are emerging and spreading globally, threatening our ability to treat common infectious diseases, and resulting in prolonged illness, disability, and death [3]. In 2019, WHO declared AMR as one of the top 10 global public health threats facing humanity [4]. Each year in the U.S., at least 2.8 million people are infected with antibiotic-resistant bacteria or fungi, and more than 35,000 people die as a result [5]. In Europe, antibiotic resistance is responsible for an estimated 33,000 deaths annually [6]. In this stage, it is an urgent requirement to discover novel effective antibiotics to respond to the challenge of antibiotic-resistant bacteria or fungi. Natural products from microorganisms have played a significant role in delivering antibiotics since the discovery of penicillin in the 1940s [7]. New microbial species in special environments and new approaches such as genome mining and high throughput chemical screening are required to exploit novel antibiotics to answer the challenge of AMR [8].

The harsh environment of Antarctica, including cold, dry and intense ultraviolet irradiation, gives the microbes unique physiological and biochemical characteristics and possesses the capacity to produce novel bioactivities and secondary metabolites [9]. In recent years, many new compounds with antibacterial or antifungal activities have been found in Antarctic fungi. For example, new polyketides ketidocillinones B and C, isolated from Antarctic sponge-derived fungus, exhibited potent antibacterial activities against *Pseudomonas aeurigenosa*, *Mycobacterium phlei*, and MRCNS (methicillin-resistant coagulase-negative *staphylococci*) [10]. For another example, new aspochracin-type cyclic tripeptides sclerotiotides M and N, deriving from Antarctic fungus, showed broad antimicrobial activities against a panel of pathogenic strains [11]. Our recent findings also discovered a new polyketide pseudophenone A, produced by an Antarctic fungus, exhibited antibacterial activities against a panel of strains [12].

To find novel antibiotics, culturable fungi from soil samples, collected from Fildes Peninsula, Antarctica, were isolated and purified, and it was found that three of the fungi crude extracts, *Chrysosporium* sp. HSXSD-11-1, *Cladosporium* sp. HSXSD-12 and *Acrostalagmus luteoalbus* CH-6, exhibited significant antifungal activities and the crude extracts of *A. luteoalbus* CH-6 showed the strongest antifungal and antibacterial activities. A series of active compounds, focusing on alkaloids and terpenoids, had been isolated from the genus of *Acrostalagmus* by previous researchers [13,14,15,16,17,18,19,20,21,22], which further proved the potential capacity of the fungus *A. luteoalbus* CH-6 to produce novel antibiotics. Molecular networking has become an outstanding method to visualize and identify the secondary metabolites of the fungi crude extracts in non-targeted MS^2^ data [23,24,25]. Hence, we performed systematic research on secondary metabolites of the fungus *A. luteoalbus* CH-6 with the strategies of bioactivity tracking and molecular networking. In this report, the bioactivity screening of the antimicrobial fungi, and the isolation, structure elucidation, and antimicrobial activity evaluations of the isolated compounds are reported and discussed.

## 2. Results and Discussion

### 2.1. Isolation and Antimicrobial Screening of Soil-Derived Fungi from Fildes Peninsula, Antarctica

Eighty-two strains of cultivable fungi were isolated from the soil samples, collected in Fildes Peninsula, Antarctica, at the Chinese 35th Antarctic expedition in 2019. According to their different community morphology (Appendix A), 26 strains were selected and fermented by two culture conditions to obtain 52 crude extracts. The antimicrobial activities of the 52 crude extracts were evaluated by one pathomycete, four pathogenic bacteria, and ten marine fouling bacteria (Table 1, Table 2 and Table 3). Three fungal crude extracts, numbered HSXSD-11-1, HSXSD-12, and CH-6, exhibited significant antifungal activity against *C. albicans* with 71.4–78.7% growth inhibition at 50 μg/mL (Table 1). Among them, the strain CH-6 displayed the strongest antimicrobial activities against *C. albicans* and *A. salmonicida* with 72.3–84.8% growth inhibition (Table 1 and Table 2).

### 2.2. Identification of the Bioactive Fungi

The three most active fungi were identified by comparing their ITS-rDNA sequences with those in the National Center for Biotechnology Information (NCBI) database, combined with their morphological characteristics. Concretely, the fungus HSXSD-11-1 was identified as *Chrysosporium* sp. whose 647 bp ITS sequence had 99.52% identity to that of *Chrysosporium* sp. 2 JC-2013 (HG329729.1) with the query coverage of 95%. The fungus HSXSD-12 was identified as *Cladosporium* sp. whose 510 bp ITS sequence had 100% identity and 97% query coverage to that of *Cladosporium* sp. CLAD127 (MK111582.1). The fungus CH-6 was identified as *Acrostalagmus*
*luteoalbus* whose 574 bp ITS sequence had 100% identity to that of *A. luteoalbus* J23B1 (MK389477.1) with the query coverage of 95%. The ITS-rDNA sequences of HSXSD-11-1, HSXSD-12, and CH-6 were submitted to GenBank and obtained the accession numbers of MT367260.1, MT367261.1, and MT367202.1, respectively.

### 2.3. Secondary Metabolites Profile Visualization and Annotation by Molecular Networking

The crude extracts of *A. luteoalbus* CH-6 were subjected to UHPLC-MS/MS analysis to obtain MS^2^ data and then the data were converted into .mzXML format to submit into Global Natural Product Social Molecular Networking (GNPS) online workflow for the molecular network of the fungus secondary metabolites profile (Figure 1). The network was visualized by Cytoscape 3.8.0. The molecular network of the fungus secondary metabolites profile contained 615 nodes and 738 edges, suggesting that 615 compounds with different molecular weights were found in the crude extracts of *A. luteoalbus* CH-6. The 21 yellow nodes were compounds characterized by molecular networking (Appendix A). The red nodes were unannotated compounds that might be new compounds according to the result of the molecular networking.

In order to identify the compounds’ structures in the first family with the maximum nodes, the literature [13,14,15,16,17,18,19,20,21,22] about secondary metabolites of *A. luteoalbus* were studied and it was found that thiodiketopiperazine derivatives were the main products of this species, and then annotated 20 analogs in the first family in the molecular network according to the previous studies about thiodiketopiperazine derivatives (Figure 1 and Figure 2) [16,20,26,27,28,29,30,31,32,33].

The *α*-pyrone derivatives (Figure 1 and Figure 3) were also annotated in the molecular network according to the literature [34,35,36,37]. The retention time of the mapped *α*-pyrones in HPLC fingerprint was pointed out with low yield suggesting that it is difficult to isolate *α*-pyrone derivatives in the extracts of the fungus *A. luteoalbus* CH-6.

### 2.4. Structure Elucidations of Isolated Compounds ***1***–***6***

The crude extracts obtained by static culture of *A. luteoalbus* CH-6 showed the strongest antifungal and antibacterial activities. Therefore, they were separated through a bioactivity-guided strategy which led to the isolation of compounds **1**–**6**. Acrostalapyrone A (**1**) (Figure 4) was obtained as an amorphous powder. Its molecular formula, C_14_H_20_O_4_, was determined by HR-ESI-MS spectrum (Appendix A), indicating five degrees of unsaturation. Careful analysis of ^1^H NMR, ^13^C NMR, and HSQC spectra (Appendix A) of 1 revealed five methyl signals, including one oxygenated methyl at δ_H_ 3.91 (3H, s), δ_C_ 56.3, four methines, including two unsaturated methines at δ_H_ 6.46 (1H, dd, 10.3, 1.5 Hz), δ_C_ 136.8 and δ_H_ 6.13 (1H, s), δ_C_ 92.4, and one oxygenated methine at δ_H_ 3.72 (1H, p, 6.4 Hz), δ_C_ 71.9, and five unsaturated quaternary carbons, including one ester group at δ_C_ 165.2 (Table 4). The five unsaturated quaternary carbons and two unsaturated methines represented four degrees of unsaturation, combined with the whole five degrees of unsaturation, provided the existence of a ring. All of these NMR spectra characters revealed that **1** was a pyrone compound. Further analyzing these data discovered that **1** was very similar to phomenin A [38] and phomapyrone E [39]. The most obvious differences in the NMR data between **1** and phomenin A were the two olefinic carbons signals in phomenin A were substituted by two methines at δ_H_ 3.72 (1H, p, 6.4 Hz), δ_C_ 71.9, and δ_H_ 2.60 (1H, dp, 10.3, 6.7 Hz), δ_C_ 41.1 in **1** (Table 4). This was further elucidated by the HMBC correlations from H-9 Me to C-10 and from H-11 to C-9, and the COSY relationships of H-8/H-9, H-9/H-9 Me, H-9/H-10 and H-10/H-11 (Figure 5). Thus the planar structure of **1** was unambiguously confirmed.

The relative configurations of **1** were determined by NOESY spectrum (Appendix A). The NOESY correlation between H-7 Me and H-9 revealed the *E* configuration of the olefinic bond at C-7 and C-8 (Figure 6). The cross-peaks of H-8/H-9 Me and H-8/H-11 in the NOESY spectrum proved H-9 Me and H-11 were in the same face. Thus, the relative configurations of **1** were elucidated to be 7*E*,9*S*,10*R* or 7*E*,9*R*,10*S.*

The absolute configurations of **1** were ascertained by modified Mosher’s methods [40,41]. The (*S*)- and (*R*)-MTPA esters of **1**, **1s**, and **1r**, were obtained after treatment of **1** with (*S*)- and (*R*)-MTPA-Cl, respectively. The 10*R* configuration of **1** was revealed by the Δδ_H(**1s**–**1r**)_ values Δδ_H-11_ = +0.07, Δδ_H-9_ = −0.05, Δδ_H-9 Me_ = −0.11, Δδ_H-8_ = −0.06 (Figure 7), following the Mosher’s rules. Therefore, the absolute configurations of **1** were confirmed to be 7*E*,9*S*,10*R*. The *α*-pyrone analogs were named by the name of the isolated fungi and the structural type of the compounds [38,39], compound **1** was named acrostalapyrone A.

Acrostalapyrone B (**2**) (Figure 4) was gained as an amorphous powder with the molecular formula of C_14_H_20_O_4_ determined by HR-ESI-MS (Appendix A), which was the same as **1**. The ^1^H and ^13^C NMR data of **2** were very similar to those of **1** (Table 4) indicating that **2** and **1** shared the same plane structure. The obvious differences between the ^13^C NMR data (Table 4) of **2** and **1** were the higher field shifts of C-7 (δ_C_ 125.7 in **2** vs. δ_C_ 126.4 in **1**) and C-11 (δ_C_ 20.8 in **2** vs. δ_C_ 21.3 in **1**) which might suggest the different configurations of C-7 and C-10 in **2** and **1**.

The relative configurations of **2** were decided by its NOESY spectrum (Appendix A). The cross peak of H-7 Me and H-9 in NOESY proved the *E* configuration of the olefinic bond at C-7 and C-8 (Figure 6) which was the same as **1**. The NOESY relationship of H-9 and H-11 indicated that H-9 and H-11 were on the same side (Figure 6). The NOESY cross-peaks of H-9 Me/H-10 revealed they were on the same side. Thus, the relative configurations of **2** were decided to be 7*E*,9*S*,10*S* or 7*E*,9*R*,10*R*.

The absolute configurations of **2** were confirmed by modified Mosher’s methods [40,41]. Compound **2** reacted with (*S*)- and (*R*)-MTPA-Cl, respectively, to obtain (*S*)- and (*R*)-MTPA esters of **2**, **2s**, and **2r**. According to the Mosher’s rules, the Δδ_H(**2s**–**2r**)_ values Δδ_H-11_ = −0.05, Δδ_H-9_ = +0.01, Δδ_H-9 Me_ = +0.08, Δδ_H-8_ = +0.09 (Figure 7) confirmed the 10*S* configuration of **2**. Thus, the absolute configurations of **2** were unquestionably ascertained to be 7*E*,9*S*,10*S,* and named acrostalapyrone B.

The optical rotation (OR) values of compounds **1** (${\left[\alpha \right]}_{D}^{20}$ −33 (*c* 0.067, CH_2_Cl_2_)) and **2** (${\left[\alpha \right]}_{D}^{20}$ −48 (*c* 0.067, CH_2_Cl_2_)) are similar, therefore the absolute configurations of these two compounds cannot be elucidated by OR.

The NMR and OR data of compounds **3**–**6** (Appendix A) were exactly the same as those in the literature [19,32,33,42,43], so **3**–**6** were elucidated to be multiforisin G [42], luteoalbusin A [19,43], gliocladine C [32], and T988 C [33], respectively.

### 2.5. Antifungal and Antibacterial Activity Evaluations of Isolated Compounds

All the isolated compounds (**1**–**6**) were evaluated for their antifungal and antibacterial activities against one pathomycete *C. albicans*, four pathogenic bacteria *E. coli*, *S. aureus*, *B. subtilis*, *P. aeruginosa*, and ten marine fouling bacteria *P. fulva*, *P. aeruginosa*, *A. salmonicida*, *A. hydrophila*, *V. anguillarum*, *V. harveyi*, *P. halotolerans*, *P. angustum*, *E. cloacae,* and *E. hormaechei*. *α*-Pyrones **1**–**3** displayed no obvious antimicrobial activities against all the tested strains, and indole diketopiperazines **4**–**6** showed antimicrobial activities against a panel of strains (Table 5). Among them, compound **6** exhibited broad-spectrum antimicrobial activities against *C. albicans*, *A. salmonicida*, *P. halotolerans*, *P. fulva,* and *S. aureus* with the MIC values range from 3.125 μM to 50 μM (Table 5). Furthermore, compound **6** displayed antibacterial activity two times higher than that of the positive drug Ciprofloxacin against *A. salmonicida*. Compounds **4** and **6** showed significant antimicrobial activities against *C. albicans* and *A. salmonicida* with the MIC values of 3.125–12.5 μM (Table 5).

## 3. Materials and Methods

### 3.1. General Experimental Procedures

Polymerase chain reaction (PCR) was performed using Thermo T-100 (Thermo Fisher Scientific, Bremen, Germany). Optical rotations were tested on a JASCO P-1020 digital polarimeter (JASCO, Tokyo, Japan). The UV spectrum was recorded using an Implen Gmbh NanoPhotometer N50 Touch (Implen, Munich, Germany). NMR spectra were measured on a Bruker AVANCE NEO (Bruker, Switzerland) at 600 MHz for ^1^H and 150 MHz for ^13^C in CDCl_3_. Chemical shifts δ were recorded in ppm, using TMS as the internal standard. HR-MS spectra were measured on a Thermo Scientific LTQ Orbitrap XL spectrometer (Thermo Fisher Scientific, Bremen, Germany). HPLC separation was performed using a Hitachi Primaide Organizer Semi-HPLC system (Hitachi High Technologies, Tokyo, Japan) coupled with a Hitachi Primaide 1430 photodiode array (PDA) detector (Hitachi High Technologies, Tokyo, Japan). A Kromasil C_18_ semi-preparative HPLC column (250 × 10 mm, 5 µm) (Eka Nobel, Bohus, Sweden) was used. Silica gel (200–300 mesh; Qingdao Marine Chemical Group Co., Qingdao, China) and Sephadex LH-20 (Amersham Biosciences Inc., Piscataway, NJ, USA) were used for column chromatography. Thin-layer chromatography was performed on precoated silica gel GF254 plates (Yantai Zifu Chemical Group Co., Yantai, China).

### 3.2. Isolation and Fermentation of Soil-Derived Fungi from the Fildes Peninsula, Antarctica 

The soil samples were collected in ice-free areas (about 10 cm from the surface) of the Fields Peninsula (S62°12′, W58°58′) using sterile spatulas and sterilized WhirlPak bags (Sigma-Aldrich, St. Louis, MO, USA), and were transported to the lab in sealed foam package filled with dry ice by airplane, at the Chinese 35th Antarctic expedition in 2019 [44]. The soil samples were incubated in a water bath at 16 °C for 3 min to thaw quickly. In aseptic conditions, 10 g soil sample was mixed thoroughly in 10 mL of sterile distilled water and stood overnight to obtain a bacterial and fungal suspension. The suspension was diluted into 10^−1^, 10^−2^, and 10^−3^ with sterile distilled water, and 100 µL of each dilution along with a stock solution was transferred to PDA culture media and evenly dispersed, respectively, each strength of the suspension was repeated three times to incubate at 4 °C, 16 °C, and 28 °C for one to three weeks until no new colonies appear.

Single colonies of fungi were carefully picked into new PDA culture media repeatedly until only one colony grew in the medium. The purified fungi were transferred into cryogenic vials containing potato dextrose water (PDW) culture media with glycerol protection (*v*/*v* = 3:1), stored at −80 °C in the State Key Laboratory of Microbial Technology, Institute of Microbial Technology, Shandong University, Qingdao, China. The isolated fungi were selected with different morphological colonies and cultivated in PDW culture media in two Erlenmeyer flasks (300 mL in each 500 mL flask) at 16 °C, one in static (45 days) condition, another one in shock (14 days) condition.

### 3.3. Extraction and Bioactivities Screening of Fermented Fungi

Each of the fungal fermented culture broth (300 mL) was filtered by two layers of gauze to separate the mycelia from the broth. The mycelia were extracted three times with EtOAc (3 × 200 mL) and then repeatedly extracted with CH_2_Cl_2_–MeOH (*v*/*v*, 1:1) three times (3 × 200 mL). The broth was extracted repeatedly with EtOAc (3 × 300 mL) to get the EtOAc layer. All the extracts were combined and then evaporated to dryness under reduced pressure to afford residues.

The antifungal and antibacterial activities of the fungal extracts were evaluated by the conventional broth dilution assay [45,46]. One pathomycete, *Canidia albicans* (ATCC 10231), four pathogenic bacteria *Escherichia coli* (ATCC 25922), *Staphylococcus aureus* (ATCC 27154), *Bacillus subtilis* (ATCC 6633), *Pseudomonas aeruginosa* (ATCC 27853) were used. Ten marine fouling bacteria *P. fulva*, *P. aeruginosa*, *Aeromonas salmonicida*, *A. hydrophila*, *Vibrio anguillarum*, *V. harveyi*, *Photobacterium halotolerans*, *P. angustum*, *Enterobacter cloacae,* and *E. hormaechei*, isolated from a marine biofilm formed on the bottom of a boat, were also used because microorganisms defending themselves by producing antibiotics against competitive bacteria [47,48]. Cipofloxacin and Sea-nine 211 were used as a positive control, DMSO was used as a negative control.

The bioactive assays were tested in 96 well-plate. Each well contained 198 μL tested strain suspension (2–5 × 10^5^ CFU/mL in LB broth) and 2 μL fungal extract (final concentration was 50 μg/mL). Three replicates were performed. The plates were incubated at 37 °C for 24 h, then the OD values were tested at 600 nm in a microplate reader (TriStar^2^ S LB 942 Multimode Reader, Berthold Technologies, Germany). The inhibitory rates were calculated according to the following formula:Inhibition rate (%) = (OD_DMSO_ − OD_extrat_)/OD_DMSO_ × 100

### 3.4. The Identification of the Bioactive Fungal Strains

The identification of the bioactive fungal strains HSXSD-11-1, HSXSD-12, and CH-6 was conducted by the analysis of the 28S rRNA gene sequences. Each of the fresh fungal mycelium (about 1.00 mg) was dispersed in 50 μL lysis buffer for microorganism to direct PCR (Takara, Cat# 9164), and saved in a metal bath (Yooning, Hangzhou, China) at 100 °C for 30 min to extract its genomic DNA as template DNA. The PCR reactions were performed in a final volume of 50 μL, which was composed of template DNA (3 μL), ITS1 (1 μL), ITS4 (1 μL), PrimeSTAR^®^ Max DNA Polymerase (25 μL, Takara, Cat# R045A) and ultrapure water (20 μL), under the following procedures: (1) initial denaturation at 98 °C for 5 min; (2) denaturation at 98 °C for 30 s; (3) annealing at 55 °C for 30 s; (4) extension at 72 °C for 1 min; and (5) final extension at 72 °C for 10 min. Steps (2)–(4) were repeated 30 times. The PCR products were then submitted for sequencing (BGI, China) with the primers ITS1 and ITS4. The sequences of HSXSD-11-1, HSXSD-12, and CH-6 were searched in the NCBI nucleotide collection database through the BLAST program. The three fungi HSXSD-11-1, HSXSD-12, and CH-6 were identified by the BLAST results, combing with their morphological characteristics.

### 3.5. Molecular Networking

#### 3.5.1. UHPLC Parameters

Liquid chromatography performed at 30 °C was operated using an HPLC C_18_ column (Hitachi, 250 mm × 4.6 mm, 5 µm). The PDA detection was recorded from 190 to 400 nm and set wavelengths at 210 and 254 nm for peak characterization. The eluted mobile phases were MeOH (A pump) and H_2_O (B pump) with the gradient program (time (min), %A): (0.00, 5); (5.00, 5); (60.00, 100); (75.00, 100); (80.00, 5); and (90.00, 5). The mobile phases flow rate was 1.00 mL/min. The fungal extracts dissolved in methanol were kept at 20 °C and stored in an autosampler and the sample injection volume was 20 µL.

#### 3.5.2. MS^2^ Parameters

MS^2^ analyses were performed using high-resolution Q-TOF mass spectrometry (Bruker impactHD) coupled with an ESI source with the parameters as followed: positive-ion mode, capillary source voltage at 3500 V, drying-gas flow rate at 4 L/min, drying-gas temperature at 200 °C, and end plate offset voltage at 500 V. MS full scan mode was operated from *m*/*z* 50–1500 (100 ms scan time) with a resolution of 40,000 at *m*/*z* 1222.

#### 3.5.3. Molecular Network Analysis

The molecular network was created using the online workflow (https://ccms-ucsd.github.io/GNPSDocumentation/, accessed on 2 July 2020) on the GNPS website (http://gnps.ucsd.edu, accessed on 19 April 2022) [49]. The set parameters of the molecular networking were detailed and described previously [12]. The results were visualized by Cytoscape 3.8.0.

### 3.6. Extraction and Isolation of Compounds ***1***–***6*** from Acrostalagmus luteoalbus CH-6

The fungal strain *Acrostalagmus luteoalbus* CH-6 was fermented in a rice culture medium in 200 Erlenmeyer flasks (250 g rice and 350 mL water in each 1000 mL flask) at 16 °C in an air-conditioned room for 60 days. Rice culture medium was used for its eutrophy, simple preparation, low cost, convenience, and ease to obtain. On the other hand, the crude extracts of fungus *A. luteoalbus* CH-6 cultured in rice medium exhibited stronger antimicrobial activities against *C. albicans* and *A. salmonicida*. The fermented culture of *A. luteoalbus* CH-6 (50 kg) was extracted three times with EtOAc (3 × 4000 mL) and then repeatedly extracted with CH_2_Cl_2_–MeOH (*v*/*v*, 1:1) three times (3 × 4000 mL). All the extracts were combined and then evaporated to dryness under reduced pressure to afford a residue (376 g). The residue was subjected to vacuum liquid chromatography (VLC) on silica gel using step gradient elution with EtOAc–petroleum ether (PE) (0–100%) and then with MeOH–EtOAc (0–100%) to afford eight fractions (Fr.1–Fr.8). All the eight fractions (Fr.1–Fr.8) were evaluated for their antimicrobial activities against *C. albicans* and *A. salmonicida* and the results showed that the primary bioactive compounds produced by the fungus *A. luteoalbus* CH-6 were focused on Fr.3 and Fr.4. Through the analysis of HPLC-DAD-UV fingerprints (Appendix A) and antimicrobial activity evaluations of Fr.1–Fr.8 (Appendix A), Fr.3, and Fr.4 were combined together into new Fr.3 for their similar HPLC-DAD-UV fingerprints and potent antimicrobial activities. Fr.3 was subjected to silica gel chromatographic column (CC) eluting with EtOAc–PE (0–100%) to give eight fractions (Fr.3.1–Fr.3.8). Among them, Fr.3.5 and Fr.3.6 displayed strong antimicrobial activities (Appendix A). Fr.3.5 was isolated by CC on Sephadex LH-20 eluted with CH_2_Cl_2_–MeOH (v/v, 1:1) to afford four fractions (Fr.3.5.1–Fr.3.5.4). The bioactivity test results of Fr.3.5.1–Fr.3.5.4 exhibited Fr.3.5.2 and Fr.3.5.3 with significant antimicrobial activities (Appendix A). Fr.3.5.2 was purified by using semi-preparative HPLC on an ODS column (Kromasil C_18_, 250 × 10 mm, 5 µm, 2 mL/min) eluted with 70% MeOH–H_2_O to give compounds **1** (3.8 mg) and **2** (3.9 mg). Fr.3.5.3 was first subjected to Sephadex LH-20 CC and further purified by HPLC eluting with 80% MeOH–H_2_O to gain **4** (7.8 mg) and **5** (29.2 mg). Fr.3.6 was separated on Sephadex LH-20 CC eluted with CH_2_Cl_2_–MeOH (v/v, 1:1) to afford four fractions (Fr.3.6.1–Fr.3.6.4). Fr.3.6.1 showed obvious antimicrobial activities (Appendix A) and was further purified by HPLC eluted with 65% MeOH–H_2_O to give compounds **3** (4.1 mg) and **6** (10.9 mg).

Acrostalapyrone A (**1**): amorphous powder; ${\left[\alpha \right]}_{D}^{20}$ −33 (*c* 0.067, CH_2_Cl_2_); UV (CH_2_Cl_2_) *λ*_max_ (log *ε*) 236 (4.86), 332 (4.74) nm; ^1^H and ^13^C NMR data, see Table 4; HR-ESI-MS *m*/*z* 253.1432 [M + H]^+^ (calcd. for C_14_H_21_O_4_, 253.1434), 275.1250 [M + Na]^+^(calcd. for C_14_H_20_O_4_Na, 275.1254).

Acrostalapyrone B (**2**): amorphous powder; ${\left[\alpha \right]}_{D}^{20}$ −48 (*c* 0.067, CH_2_Cl_2_); UV (CH_2_Cl_2_) *λ*_max_ (log *ε*) 234 (4.67), 333 (4.48) nm; ^1^H and ^13^C NMR data, see Table 4; HR-ESI-MS *m*/*z* 253.1434 [M + H]^+^ (calcd. for C_14_H_21_O_4_, 253.1434), 275.1252 [M + Na]^+^(calcd. for C_14_H_20_O_4_Na, 275.1254).

### 3.7. Preparation of the (S)- and (R)-MTPA Esters of ***1*** and ***2***

(*S*)-(−)-*α*-methoxy-*α*-(trifluoromethyl)phenylacetyl chloride ((*S*)-MTPA-Cl) (10 μL) was added to a stirred pyridine solution (500 μL) of **1** (1.0 mg) and 4-(dimethylamino)pyridine (2.0 mg). The mixture was stirred at rt for 10 h. The reaction mixture was evaporated to dryness under reduced pressure to get the reaction product, and then purified by HPLC eluted with 80% MeOH–H_2_O to give (*S*)-MTPA ester **1s**. Treatment of **1** (1.0 mg) with (*R*)-MTPA-Cl (10 μL) as described above yielded the corresponding (*R*)-MTPA ester **1r**. By the same procedure as for the preparation of the (*S*)- and (*R*)-MTPA esters of **1**, (*S*)-MTPA ester (**2s**) and (*R*)-MTPA ester (**2r**) of **2** were obtained.

(*S*)-MTPA ester of **1** (**1s**): ^1^H NMR (600 MHz, CDCl_3_) *δ* 7.53 (dd, *J* = 6.8, 2.9 Hz, 2H, aromatic protons), 7.43–7.37 (m, 3H, aromatic protons), 6.34 (dd, *J* = 10.2, 1.4 Hz, 1H, H-8), 6.11 (s, 1H, H-5), 5.10–5.05 (m, 1H, H-10), 3.90 (s, 3H, 4-OMe), 3.57 (d, *J* = 1.3 Hz, 3H, OMe-MTPA), 2.79 (dq, *J* = 10.2, 6.8 Hz, 1H, H-9), 1.94 (s, 3H, 3-Me), 1.84 (d, *J* = 1.4 Hz, 3H, 7-Me), 1.32 (d, *J* = 6.3 Hz, 3H, H-11), 0.95 (d, *J* = 6.8 Hz, 3H, 9-Me); HR-APCI-MS *m*/*z* 469.1828 [M + H]^+^ (calcd. for C_24_H_26_O_6_F_3_, 469.1833).

(*R*)-MTPA ester of **1** (**1r**): ^1^H NMR (600 MHz, CDCl_3_) *δ* 7.52 (dd, *J* = 6.8, 3.0 Hz, 2H, aromatic protons), 7.44–7.38 (m, 3H, aromatic protons), 6.40 (dd, *J* = 10.2, 1.4 Hz, 1H, H-8), 6.12 (s, 1H, H-5), 5.08–5.04 (m, 1H, H-10), 3.90 (s, 3H, 4-OMe), 3.52 (d, *J* = 1.2 Hz, 3H, OMe-MTPA), 2.84 (dq, *J* = 10.2, 6.7 Hz, 1H, H-9), 1.95 (s, 3H, 3-Me), 1.89 (d, *J* = 1.4 Hz, 3H, 7-Me), 1.25 (d, *J* = 6.3, 3H, H-11), 1.06 (d, *J* = 6.7 Hz, 3H, 9-Me); HR-APCI-MS *m*/*z* 469.1826 [M + H]^+^ (calcd. for C_24_H_26_O_6_F_3_, 469.1833).

*δ***_1s_**–*δ***_1r_**: H-8 = −0.06, H-5 = −0.01, H-10 = +0.02, H-9 = −0.05, H-3Me = −0.01, H-7Me = −0.05, H-11 = +0.07, H-9Me = −0.11.

(*S*)-MTPA ester of **2** (**2s**): ^1^H NMR (600 MHz, CDCl_3_) *δ* 7.49–7.44 (m, 2H, aromatic protons), 7.37–7.30 (m, 3H, aromatic protons), 6.36 (dd, *J* = 10.0, 1.4 Hz, 1H, H-8), 6.09 (s, 1H, H-5), 5.03 (p, *J* = 6.4 Hz, 1H, H-10), 3.89 (s, 3H, 4-OMe), 3.47 (d, *J* = 1.2 Hz, 3H, OMe-MTPA), 2.85 (dq, *J* = 10.0, 6.8 Hz, 1H, H-9), 1.96 (s, 3H, 3-Me), 1.86 (d, *J* = 1.4 Hz, 3H, 7-Me), 1.30 (d, *J* = 6.4 Hz, 3H, H-11), 1.07 (d, *J* = 6.8 Hz, 3H, 9-Me); HR-APCI-MS *m*/*z* 469.1828 [M + H]^+^ (calcd. for C_24_H_26_O_6_F_3_, 469.1833).

(*R*)-MTPA ester of **2** (**2r**): ^1^H NMR (600 MHz, CDCl_3_) *δ* 7.48 (dd, *J* = 6.7, 2.9 Hz, 2H, aromatic protons), 7.35–7.30 (m, 3H, aromatic protons), 6.27 (dd, *J* = 9.9, 1.4 Hz, 1H, H-8), 6.02 (s, 1H, H-5), 5.04 (p, *J* = 6.4 Hz, 1H, H-10), 3.88 (s, 3H, 4-OMe), 3.56 (d, *J* = 1.3 Hz, 3H, OMe-MTPA), 2.84 (dq, *J* = 9.9, 6.8 Hz, 1H, H-9), 1.95 (s, 3H, 3-Me), 1.81 (d, *J* = 1.4 Hz, 3H, 7-Me), 1.35 (d, *J* = 6.4 Hz, 3H, H-11), 0.99 (d, *J* = 6.8 Hz, 3H, 9-Me); HR-APCI-MS *m*/*z* 469.1830 [M + H]^+^ (calcd. for C_24_H_26_O_6_F_3_, 469.1833).

*δ***_2s_**–*δ***_2r_**: H-8 = +0.09, H-5 = +0.07, H-10 = −0.01, 4-OMe = +0.01, H-9 = +0.01, H-3Me = +0.01, H-7Me = +0.05, H-11 = −0.05, H-9Me = +0.08.

### 3.8. Antibacterial and Antifungal Activity Evaluations of the Isolated Compounds ***1***–***6***

The preliminary screening method of antibacterial and antifungal activities of **1**–**6** was the same as those of fungal extracts. The MIC values of some active target compounds were evaluated using the 2-fold serial-dilution method. The concentrations of the compounds ranged from 100 µM to 0.78125 µM. The other steps were the same as the method of primary screening.

## 4. Conclusions

In summary, three antimicrobial fungi, *Chrysosporium* sp. HSXSD-11-1, *Cladosporium* sp. HSXSD-12, and *Acrostalagmus luteoalbus* CH-6, were discovered from the soil samples of the Fildes Peninsula, Antarctica. Bioassay-guided searching of antimicrobial secondary metabolites of *A. luteoalbus* CH-6, combined with molecular networking, led to the isolation of two new *α*-pyrones, acrostalapyrones A (**1**) and B (**2**), and one known analog, multiforisin G (**3**), as well as three known indole diketopiperazines, luteoalbusin A (**4**), gliocladine C (**5**), and T988 C (**6**). Compounds **4** and **6** showed significant antimicrobial activities against *C. albicans* and *A. salmonicida*. In particular, the antibacterial activity against *A. salmonicida* of **6** displayed two times higher than that of the positive drug Ciprofloxacin. This is the first time to find *α*-pyrones in the fungal genus *Acrostalagmus* and the first report to discover significant antimicrobial activities of compounds **4** and **6** against *C. albicans* and *A. salmonicida*. This study further demonstrates the great potential of Antarctic fungi in the development of new compounds and antibiotics.

## Figures and Tables

**Figure 1 marinedrugs-20-00334-f001:**
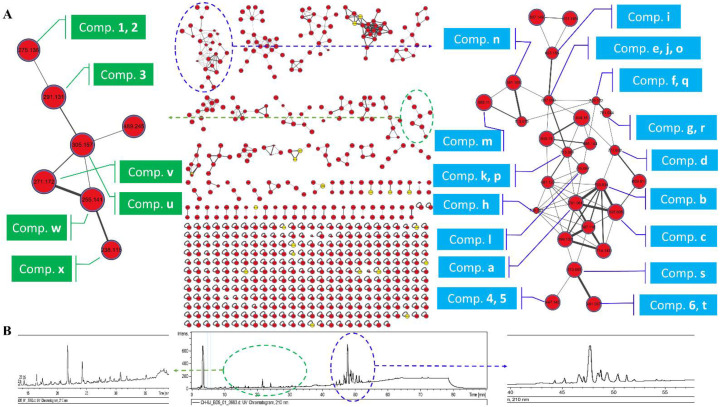
Molecular network of the fungus *A. luteoalbus* CH-6. (**A**) A full scan of the secondary metabolites’ profiles molecular network of the fungus *A. luteoalbus* CH-6 and annotated compounds. (**B**) HPLC fingerprint of the secondary metabolites of the fungus *A. luteoalbus* CH-6.

**Figure 2 marinedrugs-20-00334-f002:**
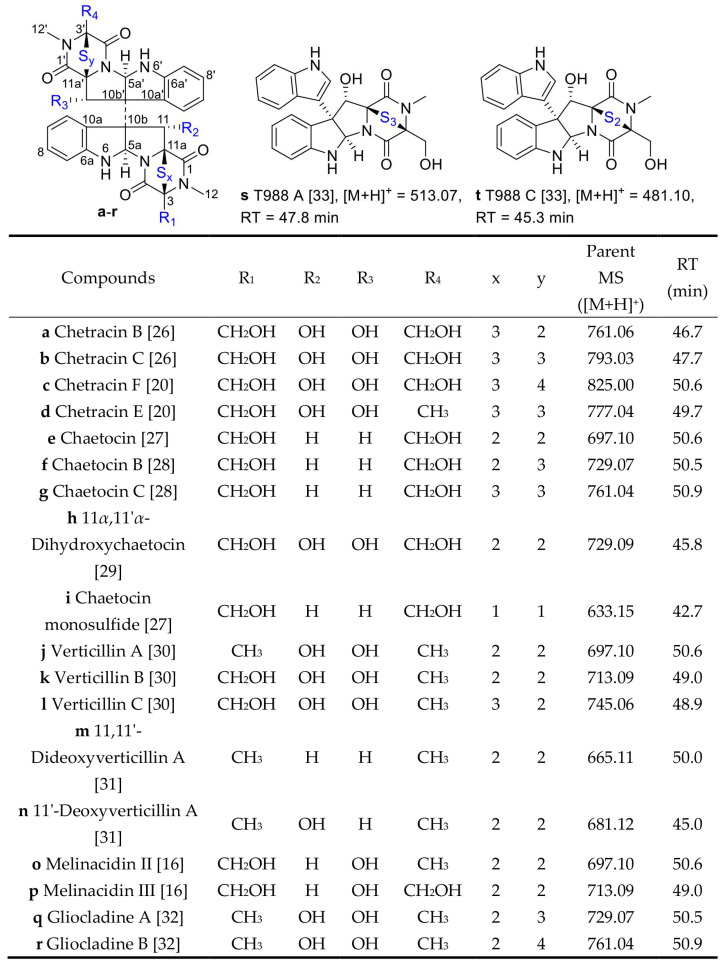
Structures of thiodiketopiperazine derivatives **a**−**t** and their retention time in HPLC [16,20,26,27,28,29,30,31,32,33].

**Figure 3 marinedrugs-20-00334-f003:**
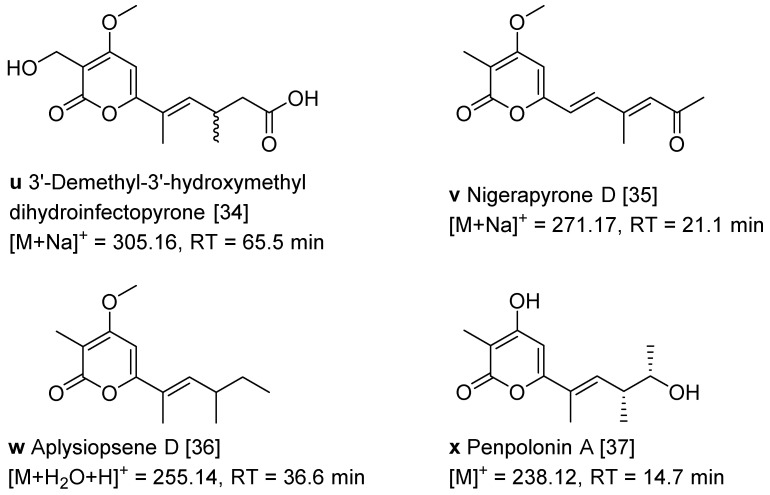
Structures of *α*-pyrone derivatives **u**−**x** and their retention time in HPLC [34,35,36,37].

**Figure 4 marinedrugs-20-00334-f004:**
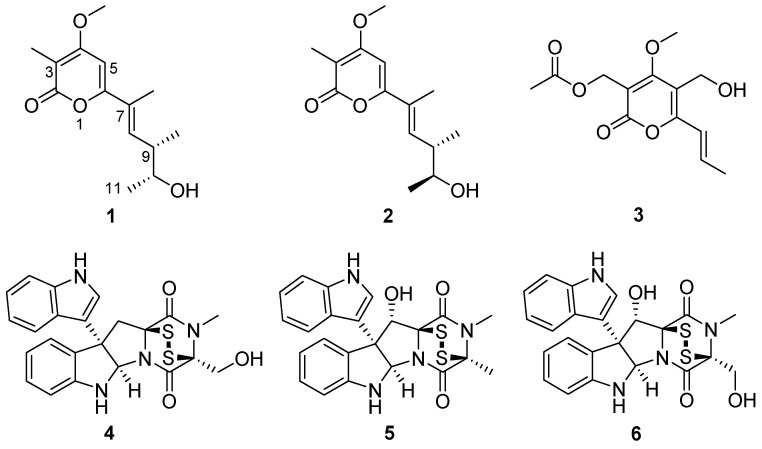
Structures of compounds **1**−**6**.

**Figure 5 marinedrugs-20-00334-f005:**
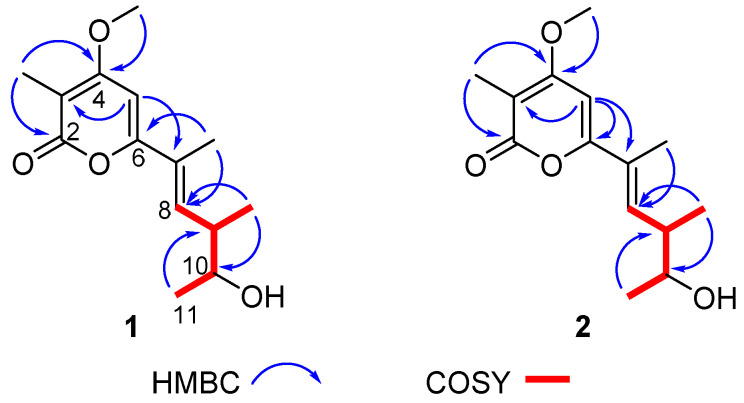
Key COSY and HMBC correlations of compounds **1** and **2**.

**Figure 6 marinedrugs-20-00334-f006:**
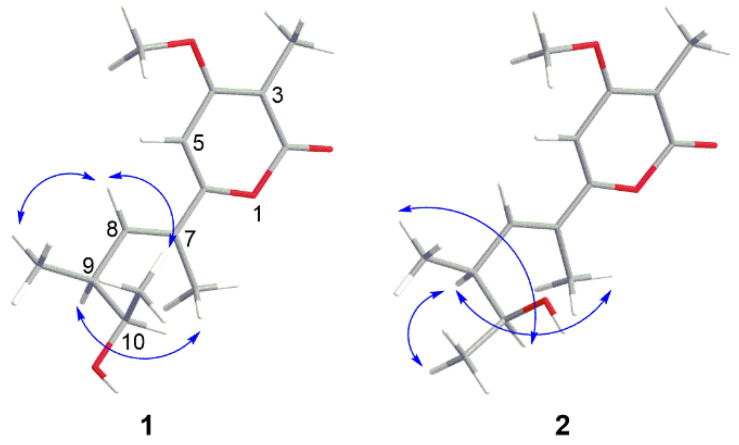
Key NOESY correlations of compounds **1** and **2**.

**Figure 7 marinedrugs-20-00334-f007:**
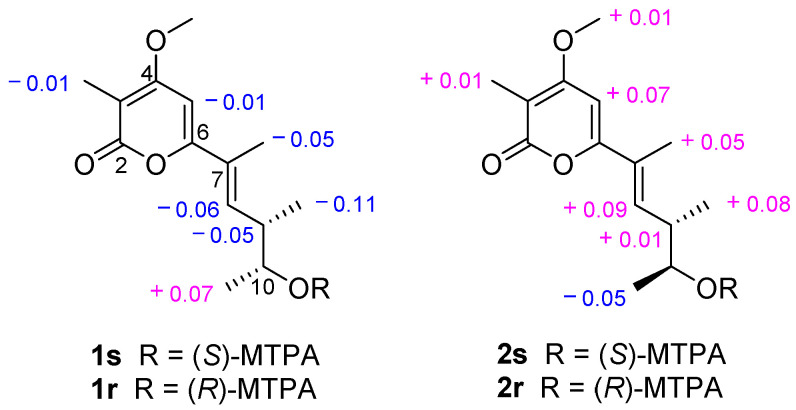
Values of ∆δ_H(*S*-*R*)_ (measured in CDCl_3_) of the MTPA esters of **1** and **2**.

**Table 1 marinedrugs-20-00334-t001:** Antimicrobial activities of the culture filtrate extracts (50 μg/mL) of the Antarctic-soil-derived fungi against one pathomycete and four pathogenic bacteria.

Strain Number	Inhibition Rate % ^a^ (Staticity/Shaking)				
	*C. albicans*	*E. coli*	*S. aureus*	*B. subtilis*	*P. aeruginosa*
HSXSD-11-1	75.0 ± 3.9/71.4 ± 0.7	–/15.2 ± 4.2	41.4 ± 2.6/40.3 ± 1.8	13.6 ± 0.5/18.0 ± 4.6	–/–
HSXSD-12	76.0 ± 3.1/–	–/–	–/25.1 ± 0.6	–/12.9 ± 2.8	–/–
HSX2#-13	–/–	–/–	–/–	–/16.6 ± 2.5	–/–
HSX6#-16	–/18.9 ± 4.1	–/–	–/24.0 ± 1.5	15.7 ± 4.8/21.4 ± 1.0	10.4 ± 2.4/11.9 ± 4.2
CH-6	78.7 ± 1.2/72.3 ± 0.8	–/–	39.3 ± 3.7/49.6 ± 4.1	19.2 ± 2.4/17.8 ± 0.3	12.8 ± 4.0/16.1 ± 4.6
CH-9	–/–	–/–	–/–	12.2 ± 2.1/17.1 ± 0.5	–/–
DLW-10	–/12.2 ± 4.9	17.1 ± 1.3/17.6 ± 3.0	20.6 ± 4.5/24.4 ± 1.1	–/12.3 ± 2.8	–/–
WLG-10	–/–	21.5 ± 2.7/–	16.9 ± 2.3/–	–/–	–/–
HSXSD-6	–/–	–/14.1 ± 2.4	–/–	–/12.8 ± 3.2	–/–
HSXSD-10	–/–	13.8 ± 1.0/16.6 ± 1.9	–/22.5 ± 0.1	–/15.3 ± 0.8	–/–
HSX2#-5	–/–	22.2 ± 2.4/–	–/16.4 ± 0.7	–/18.7 ± 1.3	–/12.3 ± 2.8
HSX2#-7	–/–	10.2 ± 2.6/–	–/13.8 ± 1.5	–/14.4 ± 1.7	–/17.1 ± 2.0
HSX2#-11	29.6 ± 4.5/14.3 ± 3.6	–/–	19.6 ± 3.5/27.3 ± 0.2	12.9 ± 1.3/18.2 ± 1.2	–/–
HSX2#-12	23.2 ± 2.4/22.7 ± 1.8	–/–	18.6 ± 0.5/27.9 ± ±0.1	14.7 ± 5.1/–	–/–
HSX2#-15	21.7 ± 6.8/–	–/–	–/12.8 ± 2.1	15.1 ± 4.5/11.5 ± 0.7	–/–
HSX11#-3	24.7 ± 4.9/–	–/12.6 ± 9.5	–/22.1 ± 0.8	16.1 ± 2.5/–	–/–
HSX11#-4	20.5 ± 3.2/–	–/–	–/–	12.8 ± 0.5/15.2 ± 1.6	–/–
HSX6#-8	–/–	11.7 ± 4.0/14.4 ± 0.8	–/26.4 ± 2.6	10.2 ± 1.1/–	–/–
HSX6#-10	30.3 ± 2.0/–	23.1 ± 2.2/–	20.8 ± 1.4/25.5 ± 0.0	17.0 ± 3.5/15.3 ± 0.9	–/–
HSX6#-14	16.1 ± 5.4/–	11.1 ± 2.8/16.0 ± 2.9	24.4 ± 5.0/–	–/–	–/–
HSX8#-6	20.6 ± 1.4/–	–/13.7 ± 0.1	22.2 ± 2.7/10.9 ± 8.8	16.3 ± 1.9/–	–/–
HSX8#-9	15.9 ± 2.5/–	11.1 ± 3.8/–	21.9 ± 0.1/–	–/–	–/–
JQW-6	24.4 ± 4.7/–	11.0 ± 1.1/–	–/–	19.4 ± 2.5/–	–/–
JQW-8	18.3 ± 4.4/–	–/–	10.5 ± 1.0/–	21.3 ± 2.3/12.7 ± 3.2	–/–
WLG-7	11.1 ± 1.4/–	–/–	–/–	11.3 ± 2.2/–	–/–
DLW-7	–/–	–/–	–/–	–/–	–/–
CH-6-rice	80.5 ± 0.9	–	48.2 ± 2.5	19.8 ± 1.6	18.1 ± 3.1
CPFX (10 μM)	66.2 ± 1.8	76.5 ± 2.3	79.1 ± 1.1	81.9 ± 3.4	70.2 ± 3.3

^a^ The inhibition rate: >60% was intense, 40–60% was media, 20–40% is low, and <20% was no activity, <10% was not shown.

**Table 2 marinedrugs-20-00334-t002:** Antimicrobial activities of the culture filtrate extracts (50 μg/mL) of the Antarctic-soil-derived fungi against five marine fouling bacteria.

Strain Number	Inhibition Rate % ^a^ (Staticity/Shaking)				
	*A. salmonicida*	*V. anguillarum*	*P. angustum*	*P. aeruginosa*	*P. halotolerans*
HSXSD-11-1	–/–	26.4 ± 2.6/32.0 ± 1.5	16.8 ± 2.3/14.6 ± 1.0	18.4 ± 4.5/14.9 ± 3.7	–/–
HSXSD-12	–/–	29.6 ± 3.5/–	16.1 ± 1.3/10.1 ± 2.0	12.1 ± 0.6/–	–/–
HSX2#-13	–/–	–/11.4 ± 2.9	16.8 ± 1.6/–	10.7 ± 0.6/–	–/–
HSX6#-16	–/–	24.5 ± 3.6/24.4 ± 0.5	17.5 ± 0.7/14.5 ± 1.1	–/–	–/–
CH-6	84.8 ± 1.4/77.6 ± 1.0	50.6 ± 2.6/58.3 ± 3.1	12.9 ± 0.8/14.4 ± 0.4	22.1 ± 2.5/13.4 ± 4.0	–/–
CH-9	–/–	10.9 ± 2.2/10.2 ± 0.2	13.2 ± 0.3/11.8 ± 0.8	–/–	–/–
DLW-10	–/–	–/20.8 ± 2.9	–/–	–/–	–/–
WLG-10	–/–	–/–	10.8 ± 0.5/–	16.8 ± 0.1/–	–/–
HSXSD-6	–/–	–/–	13.1 ± 1.7/–	–/–	–/–
HSXSD-10	–/–	–/–	–/–	–/–	–/–
HSX2#-5	–/–	–/–	–/–	–/–	–/–
HSX2#-7	–/–	–/–	12.3 ± 0.8/–	–/–	–/–
HSX2#-11	33.4 ± 2.5/13.3 ± 4.1	–/–	13.7 ± 1.2/–	–/–	–/–
HSX2#-12	–/17.4 ± 3.8	18.3 ± 0.8/–	13.7 ± 1.0/–	–/–	–/–
HSX2#-15	–/13.9 ± 3.8	19.2 ± 0.7/–	14.0 ± 0.1/–	–/–	–/–
HSX11#-3	–/–	23.8 ± 0.6/–	10.6 ± 1.0/–	–/–	–/–
HSX11#-4	–/17.9 ± 0.7	–/–	–/–	–/–	–/–
HSX6#-8	–/–	–/–	–/–	–/–	–/–
HSX6#-10	–/–	12.7 ± 1.5/–	–/–	–/–	–/–
HSX6#-14	–/–	–/–	–/14.3 ± 2.9	–/–	–/–
HSX8#-6	–/22.3 ± 4.5	–/–	–/14.5 ± 5.2	–/–	–/–
HSX8#-9	–/–	–/–	–/16.7 ± 3.0	–/–	–/–
JQW-6	13.6 ± 2.4/–	21.5 ± 3.2/–	–/16.7 ± 2.4	–/–	–/–
JQW-8	14.6 ± 2.4/–	18.2 ± 0.6/–	10.6 ± 1.5/13.3 ± 0.8	–/–	–/–
WLG-7	21.8 ± 2.1/–	12.8 ± 4.1/–	10.2 ± 0.5/12.4 ± 2.7	–/–	–/–
DLW-7	–/–	–/–	–/11.7 ± 0.9	–/–	–/–
CH-6-rice	86.7 ± 1.1	57.1 ± 2.4	12.2 ± 3.2	20.9 ± 3.1	–
Sea-nine 211 (10 μM)	78.1 ± 3.7	71.0 ± 2.1	81.5 ± 2.9	69.1 ± 2.1	67.5 ± 4.5

^a^ The inhibition rate: >60% was intense, 40–60% was media, 20–40% is low, and <20% was no activity, <10% was not shown.

**Table 3 marinedrugs-20-00334-t003:** Antimicrobial activities of the culture filtrate extracts (50 μg/mL) of the Antarctic-soil-derived fungi against another five marine fouling bacteria.

Scheme Number	Inhibition Rate % ^a^ (Staticity/Shaking)				
	*E. cloacae*	*E. hormaechei*	*P. fulva*	*V. harveyi*	*A. hydrophila*
HSXSD-11-1	–/–	12.0 ± 1.9/–	–/–	15.5 ± 1.8/12.6 ± 1.3	–/–
HSXSD-12	–/–	–/–	10.1 ± 2.2/–	16.2 ± 3.6/10.3 ± 2.9	–/–
HSX2#-13	10.5 ± 0.7/19.1 ± 0.7	10.3 ± 0.3/12.9 ± 0.5	–/12.8 ± 0.8	16.9 ± 3.2/11.7 ± 3.9	–/12.8 ± 0.7
HSX6#-16	13.3 ± 1.6/14.0 ± 1.8	11.6 ± 1.0/–	14.3 ± 1.8/10.4 ± 0.8	18.6 ± 1.6/10.5 ± 2.3	–/–
CH-6	–/–	10.1 ± 0.6/15.4 ± 0.8	11.4 ± 2.2/13.3 ± 1.2	20.0 ± 0.6/21.0 ± 0.9	19.4 ± 3.5/14.3 ± 2.3
CH-9	21.1 ± 3.2/21.0 ± 0.4	11.5 ± 2.3/14.1 ± 0.3	12.2 ± 3.2/18.8 ± 1.1	13.1 ± 3.5/15.5 ± 1.9	–/13.5 ± 1.6
DLW-10	–/–	–/–	–/11.9 ± 2.2	13.9 ± 1.8/15.5 ± 2.1	–/–
WLG-10	–/–	–/–	–/–	–/–	–/–
HSXSD-6	13.1 ± 7.5/–	–/–	–/–	13.4 ± 1.5/10.7 ± 0.7	–/–
HSXSD-10	–/–	–/–	–/–	–/11.2 ± 1.7	–/–
HSX2#-5	–/10.1 ± 1.0	–/–	–/–	–/10.1 ± 1.2	–/–
HSX2#-7	–/–	–/–	–/–	–/10.8 ± 1.8	–/–
HSX2#-11	–/–	–/–	–/–	10.8 ± 3.2/–	–/–
HSX2#-12	–/–	–/–	–/–	12.4 ± 2.9/13.3 ± 1.7	–/–
HSX2#-15	10.3 ± 4.2/–	–/–	–/–	15.3 ± 1.4/13.3 ± 1.5	–/–
HSX11#-3	15.3 ± 0.5/–	–/–	–/–	15.2 ± 1.4/–	–/–
HSX11#-4	19.0 ± 0.6/–	10.7 ± 0.6/–	10.2 ± 2.6/–	12.6 ± 1.3/11.0 ± 2.9	11.9 ± 1.1/–
HSX6#-8	–/–	–/–	–/–	–/–	–/–
HSX6#-10	–/–	–/–	–/–	–/11.2 ± 2.8	–/–
HSX6#-14	–/–	–/11.1 ± 0.7	–/–	–/–	–/–
HSX8#-6	–/–	–/–	–/–	–/13.2 ± 2.8	–/–
HSX8#-9	–/–	–/11.3 ± 1.2	–/10.6 ± 1.9	–/14.4 ± 4.3	–/–
JQW-6	13.7 ± 0.8/–	–/11.9 ± 0.4	–/10.1 ± 1.1	–/16.0 ± 4.0	–/–
JQW-8	19.1 ± 1.0/–	10.2 ± 0.7/10.5 ± 1.2	–/–	15.0 ± 1.4/15.9 ± 1.2	11.5 ± 0.5/–
WLG-7	19.8 ± 0.7/–	–/–	–/–	15.1 ± 1.8/13.7 ± 3.7	12.0 ± 0.4/–
DLW-7	–/15.2 ± 2.9	–/–	–/–	–/14.5 ± 2.5	–/–
CH-6-rice	–	12.7 ± 1.1	13.6 ± 1.8	22.3 ± 1.2	17.1 ± 2.8
Sea-nine 211 (10 μM)	70.4 ± 2.5	76.3 ± 4.6	81.1 ± 3.2	83.1 ± 2.5	69.7 ± 4.0

^a^ The inhibition rate: >60% was intense, 40–60% was media, 20–40% is low, and <20% was no activity, <10% was not shown.

**Table 4 marinedrugs-20-00334-t004:** ^1^H NMR (600 MHz) and ^13^C NMR (150 MHz) data of compounds **1** and **2** in CDCl_3_.

No.	1		2	
	*δ* _C_	*δ* _H_	*δ* _C_	*δ* _H_
2	165.2, C		165.2, C	
3	102.5, C		102.6, C	
4	166.0, C		166.0, C	
5	92.4, CH	6.13, s	92.5, CH	6.15, s
6	159.8, C		159.7, C	
7	126.4, C		125.7, C	
8	136.8, CH	6.46, dd (10.3, 1.5)	137.0, CH	6.50, dd (10.1, 1.4)
9	41.1, CH	2.60, dp (10.3, 6.7)	41.4, CH	2.57, dp (10.1, 6.8)
10	71.9, CH	3.72, p (6.4)	71.7, CH	3.69, p (6.3)
11	21.3, CH_3_	1.17, d (6.4)	20.8, CH_3_	1.22, d (6.3)
3-Me	8.8, CH_3_	1.94, s	8.8, CH_3_	1.938, s
4-OMe	56.3, CH_3_	3.91, s	56.3, CH_3_	3.91, s
7-Me	13.1, CH_3_	1.94, s	13.1, CH_3_	1.943, d (1.4)
9-Me	16.5, CH_3_	1.09, d (6.7)	16.7, CH_3_	1.04, d (6.8)

**Table 5 marinedrugs-20-00334-t005:** MIC (μM) values of compounds **1**–**6** against a panel of strains.

Strains	*C. albicans*	*A. salmonicida*	*P. halotolerans*	*P. fulva*	*S. aureus*
**1**	>50	>50	>50	>50	>50
**2**	>50	>50	>50	>50	>50
**3**	>50	>50	>50	>50	>50
**4**	12.5	12.5	>50	>50	>50
**5**	25	50	>50	>50	>50
**6**	6.25	3.125	25	25	25
**CPFX**	6.25	6.25	0.195	1.56	3.125

## Data Availability

The datasets presented in this study can be found in online repositories. The names of the repository/repositories and accession number(s) can be found below: https://www.ncbi.nlm.nih.gov/nuccore/MT367260, accessed on 21 April 2020; https://www.ncbi.nlm.nih.gov/nuccore/MT367261, accessed on 21 April 2020; https://www.ncbi.nlm.nih.gov/nuccore/MT367202, accessed on 21 April 2020.

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
