# Peer review of "Bioactivity-Guided Screening of Antimicrobial Secondary Metabolites from Antarctic Cultivable Fungus Acrostalagmus luteoalbus CH-6 Combined with Molecular Networking"

_marinedrugs, 2022, doi:10.3390/md20050334_

Round 1
Reviewer 1 Report
The authors illustrate the study on the antifungal and antibacterial activities of different Antarctic fungi. Moreover, they present the isolation and characterization of four already known compounds and two new compounds containing pyrone structure.
The work appears well organized and the new compounds properly characterized. Anyway, some parts need improvements. So, I recommend the current report for publication after a revision of the manuscript in order to clarify some aspects.
The following comments have to be considered:
- The authors determined the absolute configuration by modified Mosher’s method. It should be useful to modify Fig. 4 in order to make the differences more evident for readers. I suggest to see Fig. 5 in Bioorganic Chemistry 117 (2021) 105452, where two colors are used.
- In chapter 2.4 the authors declare: “All the isolated compounds (1–6) were evaluated for their antifungal and antibacterial activities”. However, just the results for compounds (4-6) have been reported and discussed. Might they be unactive? It should be interesting to have the results, even if they are not positive. Please, comment this point.
Author Response
Dear professor,
Thanks very much for your kind comments, we have revised our manuscript carefully according to your comments point to point. The changes in our new version of manuscript have been highlighted in red.
- The modified Mosher’s method related figure has been revised into two colors according to the reference 41.
- The antifungal and antibacterial activities of compounds 1–3 have been reported and discussed in the chapter “Antifungal and Antibacterial Activity Evaluations of Isolated Compounds”.
Reviewer 2 Report
Please see attached file

Author Response
Dear professor,
Thanks very much for your kind comments, we have revised our manuscript carefully according to your comments point to point. The changes in our new version of manuscript have been highlighted in red.
- The introduction section has been rewritten according to your suggestions.
- The “filtration” has been changed into “bioactivity screening” in line 77.
- The culture condition of the CH-6 crude extract has been stated clearly in line 146.
- The word has been changed into “planar” in line 165.
- The carbon numbers of compounds 1 have been added in figures 4–
- The recent reference 41 on Mosher’s method has been added in line 179.
- According to the α-pyrone analogues references 38 and 39, the new α-pyrones were named by the name of the isolated fungi and the structural type of the compounds, so compounds 1 and 2 were named as acrostalapyrones A and B, respectively (line 183, 184 and 205).
- The determination of the absolute configurations of compound 2 has been revised in line 200.
- The structure elucidation of compounds 4–6 has been revised into “The NMR and OR data of compounds 3–6 (Figures S25–S32, Table S2) were exactly same with those in the literatures [19, 32, 33, 42, 43], so 3–6 were elucidated to be multiforisin G [42], luteoalbusin A [19, 43], gliocladine C [32], and T988 C [33], respectively” in line 209–
- The initial volume of fungal broth has been supplied in line 265. The fungal mycelia were big enough to use gauze to filtrate, so we didn’t use the method of centrifugation.
- The bioactivity-guided strategy has been reflected at chapter 3.6, and the antimicrobial activity results of the separated fractions have been supplied in Table S3.
- The “The culture (50 Kg)” has been changed into “The fermented culture of luteoalbus CH-6 (50 Kg)” in line 328.
- The optical rotation values of compounds 1 and 2 are similar, so the absolute configurations of these two compounds can’t be elucidated by OR. This comment has been supplied in line 206–
- The conclusion has been rewrite to be more concise.
Thanks again!
Sincerely,
Yours,
Ting Shi

Author Response
Dear professor,
Thanks very much for your kind comments, we have revised our manuscript carefully according to your comments point to point. The changes in our new version of manuscript have been highlighted in red.
- The molecular networking has been applied to map out the related analogs about indole diketopiperazines and α-pyrones of the fungus luteoalbus CH-6 (Figures 1–3).
- The detailed collection information of the soil samples has been supplied in chapter 3.2.
- The structures of compounds 4–6 have been redrawn according to the reference 43 (Org. Lett. 2015, 17, 17, 4268–4271).
- The paper “Phytochemistry 66 (2005) 81–87” has been cited as reference 39.
- The carbon numbers of the Figures 4–6 have been labeled.
- The MIC values have been changed in Table 5 according to the reference 46 (J Antimicrob Chemother. 2001, 48 Suppl 1:5-16.).
Thanks again for your efforts!
Sincerely,
Yours,
Ting Shi
Round 2
Reviewer 3 Report
The authors should re-write the title. The title in this version indicates the universal molecular networking-based metabolome analysis and bioactivity screening of Antarctic cultivable fungi. However, only strain CH-6 was investigated by molecular networking-based metabolome analysis. Please provide high resolution figure 1.
Author Response
Dear Editor,
Thanks very much for your effort for our manuscript “marinedrugs-1713083”. We have revised our manuscript carefully according to your comments, and revisions made to the manuscript has been marked up using the “Track Changes” function.
The title of the manuscript has been changed into “Bioactivity-guided Screening of Antimicrobial Secondary Metabolites from Antarctic Cultivable Fungus Acrostalagmus luteoalbus CH-6 Combined with Molecular Networking”.
The high resolusion Figure 1 has been supplied.
With best regards,
Yours sincerely,
Ting Shi